# The Combination of Oncolytic Virus and Antibody Blockade of TGF-β Enhances the Efficacy of αvβ6-Targeting CAR T Cells Against Pancreatic Cancer in an Immunocompetent Model

**DOI:** 10.3390/cancers17091534

**Published:** 2025-04-30

**Authors:** Zuoyi Zhao, Lauren C. Cutmore, Renato B. Baleeiro, Joseph J. Hartlebury, Nicholas Brown, Louisa Chard-Dunmall, Nicholas Lemoine, Yaohe Wang, John F. Marshall

**Affiliations:** Barts Cancer Institute, Queen Mary University of London, London EC1M 6BQ, UK; zuoyizhao@mednet.ucla.edu (Z.Z.); lauren.cutmore@nih.gov (L.C.C.); r.baleeiro@qmul.ac.uk (R.B.B.); j.hartlebury@qmul.ac.uk (J.J.H.); nfbrown@gmail.com (N.B.); l.chard@qmul.ac.uk (L.C.-D.); nick.lemoine@qmul.ac.uk (N.L.)

**Keywords:** CAR T cell therapy, pancreatic cancer, oncolytic virus, integrin, αvβ6

## Abstract

As a revolutionary treatment for malignant diseases, CAR T cell therapy has shown remarkable success; however, its application in solid tumours has long remained a major challenge in the field. In this study, we first developed a preclinical CAR T cell therapeutic model for pancreatic cancer in an immunocompetent mouse model, targeting the mouse αvβ6 integrin. To enable large-scale in vivo studies, we then introduced a straightforward and efficient method for the production and cryopreservation of mouse CAR T cells. Building on this, we subsequently established an innovative combination strategy to enhance the efficacy of CAR T cells in pancreatic cancer. By combining CAR T cell therapy with oncolytic vaccinia virus and TGF-β blockade, we observed significantly improved T cell infiltration at both primary and metastatic tumour sites, leading to enhanced tumour suppression, significantly improved overall survival and a marked reduction in metastatic burden. Thus, this study provides a practical methodology for studying CAR T cells in immunocompetent models and presents a novel preclinical combination therapeutic strategy for enhanced efficacy of CAR T cell therapy in pancreatic cancer.

## 1. Introduction

Pancreatic ductal adenocarcinoma (PDAC) ranks as the fourth leading cause of death among solid cancers and is characterized by a dismal 5-year survival rate of less than 5% [1]. Typically diagnosed at an advanced stage, PDAC often presents with widespread metastases [2,3]. Despite advancements in the treatment of many other types of cancers, survival rates for pancreatic cancer have remained stagnant over the past four decades [3], underscoring the urgent need for novel therapeutic regimes. Chimeric antigen receptor-engineered (CAR) T cells have achieved remarkable success in treating B-cell malignancies [4], yet this success has not been fully translated to the field of solid tumours [5,6]. In this study, a novel PDAC-targeting murine CAR T cell was established, and the application of CAR T cells in solid tumours was investigated and enhanced with an innovative combination therapeutic strategy.

One of the challenges in developing CAR T cells for the treatment of solid tumours is identifying target antigens that are highly expressed on the malignant cells but have limited or no expression in healthy tissues to prevent potentially lethal on-target-off-tumour toxicity [7]. Integrin αvβ6, an epithelial-specific heterodimeric transmembrane receptor, plays crucial roles in various tumorigenesis processes such as adhesion, migration and invasion [8]. Its overexpression has been observed in most PDAC tumours and matched metastases [8,9] and higher expression levels of αvβ6 have been associated with poorer survival in patients with PDAC, along with many other types of cancer [10,11,12]. Importantly, integrin αvβ6 is expressed at very low or undetectable levels in most normal adult tissues, including normal pancreas [13]. Previous studies have demonstrated the potential of integrin αvβ6 as a target for anti-cancer therapy [14,15,16,17]. A20FMDV2 (A20) is a 20-amino acid peptide derived from the foot-and-mouth disease virus that exhibits over 1000-fold greater specificity for αvβ6 compared to other integrin heterodimers [18,19]. A second-generation human CAR T cell armed with an αvβ6-specific A20 peptide binding domain was developed. The A20 CAR T cells demonstrated impressive anti-tumour effects both in vitro and in vivo in a range of solid tumour models with human tumour xenografts in immune-deficient mice by infusion of human T cells [15,20]. These models do not reflect the main challenges of developing effective CAR T therapy for solid tumours, such as the inhibitory signals of endogenous T cells within the tumours and uncertain homing ability of engineered T cells. Therefore, translating these findings into clinical success has been poor. To develop a better CAR T cell therapy for solid tumours, it is key to use a suitable preclinical model that accurately represents the intricate immune landscape of tumours [21]. CAR T cells are routinely evaluated in immunodeficient models, which allow the engraftment of human tumours and human T cells [22]. However, translating these findings into clinical success has been poor. Given that PDAC is characterized by a dense and immunosuppressive tumour microenvironment (TME) that significantly hinders treatment efficacy [23], the results observed in pre-clinical immunodeficient models rarely reflect outcomes in patients [24]. In this study, a murine A20-based CAR T cell (mA20CART) was developed to target PDAC in an immunocompetent model in order to investigate and improve efficacy of the CAR T cell therapy of solid cancers.

Successful CAR T cell therapy is also limited by poor trafficking and infiltration into solid tumours, largely due to the dense matrix and immunosuppressive TME [25]. Thus, overcoming the barriers of the TME is essential for the penetration by and efficacy of CAR T cells. Oncolytic virus (OV) therapy is a widely accepted novel cancer immunotherapy modality that targets and selectively lyses only cancer cells using genetically modified viruses, which can be armed with immunoregulatory molecules to modulate the tumour-associated immune cell landscape. OVs hold great potential for enhancing CAR T cell potency in solid tumours, owing to their capacity to activate endogenous cytotoxic T cells, induce local anti-tumour immune responses, increase tumour permeability and convert immunologically ‘cold’ tumours into ‘hot’ tumours [26,27]. In the study described here, we introduced an oncolytic Vaccinia virus (OVV) armed with mouse interleukin-21 (mIL-21) [28,29] to investigate its potential for enhancing CAR T cell infiltration and therapeutic efficacy in PDAC.

The transforming growth factor beta (TGFβ) cytokine is a crucial factor contributing to the immunosuppressive TME. The activation of TGFβ has been found to correlate with the dysregulation of the tumour extracellular matrix (ECM) and the loss of tumour-targeting abilities in adaptive and innate immune cells [30,31]. In PDAC, increased T regulatory (Treg) cells in the TME contributes significantly to its immunosuppressive nature, dampening T cell function [32,33,34]. Studies have shown that Treg-derived TGFβ is the dominant signal promoting tumour-associated immunosuppression and its inhibition can restore endogenous anti-tumour T cell activity [35]. Moreover, since TGF-β promotes epithelial–mesenchymal transition (EMT) [36] and as αvβ6 is known to activate latent TGF-β, αvβ6 may promote PDAC EMT, further enhancing its invasion and migration [10,36,37]. In normal epithelial cells, TGF-β upregulates ITGB6 gene expression inducing expression of αvβ6, which, through activation of latent TGF-β, ramps up its own expression during wound healing [38,39]. Interestingly, activated Treg cells express the related integrin, αvβ8, which also activates TGF-β, promoting immune suppression [40,41]. Therefore, as a proof of principle, in addition to OVV-mIL21, we included a TGF-β blocking-antibody to target the hostile TME.

Our data show that this novel triple-therapy significantly enhanced CAR T cell infiltration, demonstrated superior anti-tumour efficacy and exhibited strong anti-metastatic properties in an immunocompetent orthotopic model of PDAC compared to CAR T cell monotherapy.

## 2. Materials and Methods

### 2.1. Cells and Cell Culture

As described in [16], TB32043mb6s2 was generated by ectopically expressing the murine β6 gene in the parental TB32043ctrl cell line, which is naturally αvβ6-negative. Both TB32043ctrl and TB32043mb6s2 cells, which expressed a luciferase/EGFP vector (44) were cultured in Dulbecco’s Modified Eagle Medium (DMEM) (Cat: 11965092, Gibco, Fisher Scientific, Loughborough, UK) supplemented with 10% foetal bovine serum (FBS) (Cat: 10500-064, Gibco). CAR T cells were cultured in Roswell Park Memorial Institute (RPMI) 1640 Medium (Cat: 11875085, Gibco) supplemented with 10% foetal bovine serum (FBS) and 1% sodium pyruvate (100 mM) (Cat: 11360070, Gibco) + 1% MEM non-essential amino acids solution (Cat: 11140050, Gibco) + 0.055mM 2-mercaproethanol (add freshly) (Cat: 21985023, Gibco). All cell lines were incubated at 37 °C and 5% (*v*/*v*) CO_2_ in 100% humidified air.

### 2.2. Cloning of a Murine αvβ6-Targeted CAR into the MSGV Gammaretroviral Vector

A DNA fragment encoding a murine αvβ6-targeted CAR (termed mA20CAR) was designed and ordered as a gBlock Gene Fragment (IDT, Iowa, USA). DNA sequence encoded the following components in-frame from the 5′ to the 3′ ends: the signal sequence of the light chain of the 1D3 antibody, a FLAGtag, the A20FMDV2 peptide, a portion of the murine CD28 molecule (with a dileucine motif changed to diglycine), and the cytoplasmic region of the murine CD3-ζ molecule in which the first and third ITAMs of the CD3-ζ molecule were inactivated. The CAR sequence was ligated into the MSGV retroviral backbone (a generous gift from Dr. James Kochenderfer, NCI, USA). Correct insertion of the DNA was confirmed via Sanger DNA Sequencing (The Genome Centre, QMUL). Details of the constructs are shown in Appendix A.

### 2.3. Murine CAR T Cells Production

Platinum-E cells were plated and transfected with Lipofectamine 3000 Reagent (Cat: L3000015, Fisher Scientific) to produce retroviruses encoding either αvβ6-targeting CAR or non-targeting CAR. The medium was replaced the following day to minimize transfection reagent toxicity. Two days post-transfection, retrovirus particles were harvested, purified and either used immediately or stored at −80 °C. Splenocytes were isolated from the spleens of female C57BL/6 mice, and T cells were negatively selected using a magnetic separation kit (Cat: 130-095-130, Miltenyi Biotec, Woking, Surrey, UK). The T cells were cultured under two activation conditions: one group was plated at a density of 1 × 10⁶ cells per well in a 12-well plate and stimulated with mouse CD3/CD28 beads (2 × 10^7^ cells/500 μL beads, Cat: 11132D, Fisher Scientific) and human IL-2 (50 U/mL, Cat: 581904, BioLegend UK, London, UK), while the other group was stimulated with Concanavalin A (ConA; 5 μg/mL, Cat: C5275, Sigma Aldrich, Gillingham, UK) and human IL-7 (1 ng/mL, Cat: 581904, BioLegend). After 24 h of activation, the T cells were transduced with retrovirus particles. The retrovirus was diluted in 500 μL of T cell culture medium and added to a 12-well plate pre-coated with 10 μg of retronectin (Cat: T100B, TaKaRa Bio UK Ltd., London, UK). The plate was centrifuged at 1000× *g* for 90 min at 32 °C with an acceleration setting of 5 and deceleration set to 0. Following centrifugation, 2 × 10^6^ T cells were added to the plate in 500 μL of medium and centrifuged at 300× *g* for 15 min at 26 °C with an acceleration setting of 5 and deceleration of 1. The final volume in each well was adjusted to 2 mL. Forty-eight hours post-transduction, the T cells were washed by centrifugation and resuspended at a concentration of 1 × 10^6^ cells/mL in T cell culture medium containing either 10 ng/mL of human IL-7 and IL-15 or 100 IU/mL of human IL-2, with cytokines replenished every two days. T cells activated with CD3/CD28 beads were expanded for five days, and the beads were removed at the time of harvest. These cells were either tested or cryopreserved for future use. T cells activated with ConA were tested two days post-transduction using flow cytometry.

### 2.4. Flow Cytometry

Approximately 200,000 cells were resuspended in 50 uL PBS and transferred to round-bottom polypropylene tubes. The resuspended cell pellet was directly labelled with 1 μL of the primary antibody (Appendix A) and incubated at 4 °C for 1 h. Following the primary antibody incubation, the cells were washed with 1 mL of PBS. Next, 50 μL of a fluorochrome-conjugated secondary antibody (e.g., Alexa dye conjugates from Fisher Scientific), diluted 1:100 in PBS, was added to the pellet and incubated at 4 °C for 30 min. Afterward, the cells were washed twice with PBS and resuspended in 250 μL of FACS buffer (1× PBS supplemented with 10% FBS, 2mM EDTA and 0.5 μg/mL DAPI; Cat: 62248, Fisher Scientific). The samples were analysed using a BD LSR Fortessa (BD Biosciences, Wokingham, UK), and the data were processed with FlowJo software (Version 10.8.1, BD Biosciences).

### 2.5. Cytotoxicity Assay of the CAR-T Cells

Target tumour cells were plated at a density of 1 × 10^5^ cells per well in 100 μL of T cell culture medium in a 96-well round-bottom plate. T cells were counted and added at the desired effector-to-target ratios (e.g., 1:0.5, 1:1, 1:5, 1:10) in 100 μL of T cell culture medium. Each condition was performed in triplicate or quadruplicate, with appropriate controls included, including target cell-only and T cell-only conditions. The plate was incubated at 37 °C for 24 h. After incubation, the cells were stained in FACS buffer containing 0.5 μL/mL Ethidium Homodimer-1 (EthD-1, Cat: E1169, Fisher Scientific) and 0.5 μL/well anti-mouse CD45 antibody (Cat: 103133, BioLegend) in the dark for 20 min. The cells were then washed twice, and cell viability was assessed by flow cytometry, detecting the endogenous EGFP signal.

### 2.6. Cytokine Release Analysis

The supernatant from the co-culture was collected, and T cells were removed by centrifugation. The Mouse IFN-γ DuoSet ELISA kit (Cat: DY485, R&D Systems, Oxford, UK) was used to measure IFN-γ levels according to the manufacturer’s instructions. Additionally, the Proteome Profiler Mouse XL Cytokine Array (Cat: ARY028, Bio Techne, Oxford, UK) was performed following the manufacturer’s protocol to analyse cytokine profiles.

### 2.7. Cytotoxicity Assay of the Virus

The Vaccinia virus used in these studies was VVTKSTC-N1L-mIL21 (OVV-mIL21), which was derived from the Lister strain smallpox vaccine, developed as previously described [28] and grown in CV1 cells. Briefly, the thymidine kinase (TK) and N1L regions of the viral genome were deleted. A murine interleukin-21 payload was expressed in the N1L region under an H5 promoter cassette. In the TK region, a mutant version of the viral B5R gene (STC) was incorporated to improve extracellular enveloped virus production [28].

Cells were seeded at 1000–2000 cells per well in 96-well plates containing DMEM supplemented with 5% FBS. The following day, 10-fold serial dilutions of OVV-mIL21 were added to the wells in sextuplicate. Wells without virus served as the 100% survival control, while wells without cells served as the 100% killing control. Cell survival was assessed 6 days post-infection using an MTS assay (Cat:G5421 Promega, Southampton, UK). The amount of virus particles required to kill 50% of the cells was shown as the EC50.

### 2.8. Viral Infection and Replication Assay

Viral replication was assessed at three time points: 24, 48 and 72 h post-infection. Cells were seeded at 200,000 cells per well in 6-well plates, with triplicate wells for each time point. Cells were infected with vaccinia virus at a multiplicity of infection (MOI) of 5 PFUs/cell. At 24, 48 and 72 h post-infection, cells were scraped and samples were collected. Following three freeze-thaw cycles, the replication samples were titrated in 96-well plates. Viral titres were determined using the TCID50 (50% tissue culture infective dose) method.

### 2.9. Animal Experiments

All experiments were approved by the UK Home Office and conducted in accordance with the guidelines for the ethical conduct in the care and use of animals. For the orthotopic model, NSG mice and C57BL/6 mice were premedicated with 100 µL of analgesia (0.03 mg/mL buprenorphine in PBS) administered subcutaneously prior to surgery. Anaesthesia was induced and maintained with isoflurane. A mid-axillary incision was made through the skin, followed by a 7–8 mm incision through the peritoneum. The spleen was exteriorized, and 30 µL of Matrigel containing 1000 TB32043mb6s2 cells was surgically injected into the head of the pancreas. The peritoneum was closed using 6-0 sutures (B. Braun, Melsungen, Germany), and the skin was closed with surgical staples, which were removed 7 days post-surgery. For the subcutaneous model, 1 × 10^6^ cells were suspended in 100 µL of PBS and injected subcutaneously into the right shoulder of female C57BL/6 mice. Tumour size was measured three times weekly using callipers. Treatments were administered as described in the Results section. Briefly, cyclophosphamide (Cat. PHR1404, Sigma Aldrich, St. Louis, MO, USA) was diluted in 150 µL of 0.9% NaCl and administered intraperitoneally at 100 mg/kg. CAR T cells and OVV-mIL21 were resuspended in 200 µL of ice-cold PBS and administered intravenously via the tail vein. Anti-TGFβ antibody (Cat. No. BE0057, Bio X Cell, Shrewsbury, UK) was administered intraperitoneally at 25 µg/kg.

### 2.10. IVIS Bioluminescence Imaging

Animals were imaged twice weekly using in-vivo luciferase imaging. Mice were anesthetised with 2% isoflurane and injected intraperitoneally with 150 mg/kg of VivoGlo™ Luciferin (Cat: P1043, Promega, UK) diluted to 15 mg/mL in PBS. Ten minutes after injection, the animals were imaged using an IVIS-200 system (PerkinElmer, Beaconsfield, UK). Living Image software (Calliper Life Sciences, Version 4.5) was used for image analysis. Luciferase activity was reported as photons per second (p/s) (examples shown in Appendix A).

### 2.11. Tissue Harvesting and Immunohistochemistry

Immediately after euthanasia, the mice were dissected and their tissues were inspected for macroscopic metastases. Selected tissues were collected and fixed in 10% formalin (Cat: BAF-0010-01A, Cellstor, Hunmanby, UK) at 4 °C overnight. After 24 h, the formalin was replaced with 70% ethanol. The tissues were subsequently processed, embedded in paraffin, sectioned into slides and stained by the Core Pathology Services at Barts Cancer Institute. The slides were scanned to include whole tissue sections, and the images were analysed using the HALO AI (Indica labs, Albuquerque, NM, USA) image analysis platform for quantification of tissue staining and immunohistochemistry.

## 3. Results

### 3.1. Murine A20FMDV2 Peptide-Based CAR T Cells Demonstrate a Specific Anti-Tumour Effect Toward Pancreatic Cancer Cells Overexpressing Integrin αvβ6

A fully murine αvβ6-targeting CAR was designed using the A20FMDV2 peptide as the binding moiety, referred to as mA20CAR. A non-targeting control CAR was also produced, lacking the peptide extracellular binding domain, designated as NTCART (Figure 1a). The intracellular domain contained a murine CD28 costimulatory domain, with a change of a dileucine (LL) motif to diglycine (GG) that has previously been shown to enhance CAR expression [42]. This co-stimulatory domain was followed by a murine CD3ζ signalling domain, containing a tyrosine to phenylalanine mutation in the first and third immunoreceptor tyrosine-based activation motifs (ITAMs), which has been shown to decrease T cell apoptosis [43] (Figure 1a). To assess the efficiency of the mA20CAR binding domain in binding to integrin αvβ6, flow cytometry was performed on HEK293T cells expressing the CAR. An anti-FLAGtag antibody was applied to detect the CAR expression on the surface of the HEK293T cells following retroviral transduction (Figure 1b). Recombinant αvβ6 labelled with a HIS-tag label specifically bound to mA20CAR-expressing cells but not NTCAR-expressing cells, demonstrating that the mA20CAR is able to bind to integrin αvβ6 (Figure 1b). Murine CAR T cells were produced by isolating splenocytes from C57BL/6 mice and transducing them with gamma retroviral particles encoding mA20CAR or NTCAR. The transduction efficiency was indicated by FLAGtag expression. CAR expression was detected on approximately 55% of the cells transduced with NTCAR (NTCART) and around 32% of the cells transduced with mA20CAR (mA20CART) relative to untransduced murine splenocytes (Figure 1c). Following splenocyte isolation, 25% of the cells were CD3+. The CD3+ cells preferentially expanded under the culture conditions and made up over 90% of the cells by day 6 post-isolation.

The antigen-specific effector function of the mA20CAR T cells was investigated in vitro using a pair of isogenic murine PDAC-derived tumour cell lines that were αVβ6-positive or -negative (as described in our previous study [16]). Both the αvβ6-negative TB32043 cells and the TB32043 cells ectopically expressing murine αvβ6 (TB32043mb6s2) co-expressed GFP-luciferase [16]. Here, we confirmed that the A20 peptide conjugated to Cyanine3 (Cy3) specifically binds to TB32043mb6s2 cells but not to TB32043 cells, demonstrating both the expression of αvβ6 and the specificity of the A20FMDV2 peptide (Figure 1d). The mA20CART cells exhibited specific cytotoxicity towards TB32043mb6s2 cells compared to NTCART cells at a 10:1 effector-to-target (E:T) ratio (Figure 1e). Antigen-specific target cell lysis was not detected with the NTCART cells until a 30:1 E:T ratio was reached (Figure 1f). This demonstrates that mA20CART cells can specifically target αvβ6-expressing murine PDAC cells and exhibit minimal off-target activity against antigen-negative targets. Moreover, CAR T cell activation measured by interferon-gamma (IFN-γ) release was assessed. Forty-eight hours after co-culture of the CAR T cells with the target cells, the mA20CART cells released high levels of IFN-γ when co-cultured with TB32043mb6s2, but not with TB32043ctrl (Figure 1f). In contrast, NTCART cells showed no increased IFN-γ production when co-cultured with either cell line (Figure 1f). This further indicates that mA20CART cells are specifically activated upon interaction with αvβ6-expressing target cells.

The anti-tumour activity of the mA20CART cells was evaluated in vivo. One of the significant barriers to CAR T cell therapy in PDAC is the dense desmoplastic stroma characteristic of the disease. We have previously demonstrated that the TB32043mb6s2 model recapitulates this stroma [16]. Firstly, the efficacy was assessed in an immunodeficient, as model described in Figure 1g. TB32043mb6s2 cells were orthotopically injected into the pancreas of NOD SCID gamma (NSG) mice. Tumours grew rapidly and were highly desmoplastic. The administration of mA20CART prolonged overall survival compared to NTCART-treated mice (Figure 1h). However, there was no significant difference in tumour volume between treatment groups, as measured by bioluminescence using the in vivo imaging system (IVIS) (Figure 1i). Next, we proceeded to evaluate the therapeutic efficacy in the immunocompetent model. Compared to NTCART cells, mA20CART cell treatment led to improved overall survival in the C57BL/6 mice bearing orthotopic TB32043mb6s2 PDAC tumours (Figure 1j). Nonetheless, no therapeutic effect was seen on growth of the primary tumour (Figure 1k). Notably, the improvement in survival was less pronounced compared to the immunocompromised model, suggesting that the host immune system exerts an inhibitory effect on the anti-tumour efficacy of the CAR T cells. The persistence of T cells in the mice was evident from the CD3 infiltration observed in the spleen of tumour-bearing NSG mice treated with mA20CART cells; however, only limited penetration at the tumour site was observed (Figure 1l). Therefore, to develop an effective CAR T cell therapy targeting PDAC, it was crucial to further explore therapeutic strategies that can overcome both the dense desmoplastic stroma and the immunosuppressive TME.

### 3.2. An Optimized Process for the Production and Long-Term Cryopreservation Storage of Murine CAR T Cells

The mA20CART cells demonstrated a selective response to target cells expressing integrin αvβ6. To facilitate the administration of CAR T cells in large preclinical trials in immunocompetent mice, it was crucial to develop methods for generation of substantial (10^9^–10^10^) numbers of mouse CAR T cells and this required that we investigate the impact of cryopreservation storage on murine CAR T cells. In our initial studies (described in Figure 1), mouse splenocytes were activated with Concanavalin A (ConA) and stimulated with IL-2 (see Section 2). In preliminary comparative studies showed using the method of Lanitis et al. [44] that stimulated mouse splenocytes with CD3/CD28 beads, followed by interleukin-7 (IL-7) and interleukin-15 (IL-15) to enhance CAR T cell in vitro expansion, enhance their survival and stimulate a therapeutically valuable central memory phenotype. To explore the possibility of long-term storage, we incorporated a freeze-thaw (FT) cycle to evaluate the effect of cryopreservation on murine CAR T cell functions. Here, we compared the three murine CAR T cell production protocols. All methods involved the same T cell isolation and transduction process as previously reported [44]. In the first protocol, CAR T cells were stimulated with ConA plus IL-7 for 48 h and expanded for five days in the presence of IL-2. The T cells were labelled as ConA-NTT and ConA-mA20T (for the NT or mA20 CAR expressing cells, respectively). The second protocol used CD3/CD28 beads and IL-2 for the initial stimulation, followed by expansion in media containing IL-15 and IL-7. The T cells were labelled as Beads-NTT and Beads-mA20T. For the third protocol, CAR T cells produced using the second protocol were cryopreserved for a week and then thawed for further evaluations (the scheme described in Figure 2). The T cells were labelled as Beads-FT-NTT and Beads-FT-mA20T. The viability and surface expression of key markers on the CAR-expressing cells were evaluated, including CD3, CD4, CD8 and PD1 and CD62L and CD44 expression, to evaluate the central memory and effector populations within the CD8+ and CD4+ T cell subtypes, respectively.

Compared with ConA-activated T cells, CD3/CD28 bead stimulation resulted in a higher T cell viability (increased by approximately 50%) and transduction efficiency (increased by approximately 12%), as indicated by Ethidium Homodimer-1 (EthD-1) and FLAG-tag positivity (Figure 3a). The mA20CAR-expressing T cells activated by beads (Beads-mA20T) exhibited approximately 32% lower frequency of PD1-positive cells compared to the ConA group (ConA-mA20T), indicating decreased exhaustion [45]. The Beads-mA20T exhibited a greater proportion of central memory CAR T cells (CARTcm) compared to effector CAR T cells, particularly within the CD8+ CAR T cell population, accounting for 81% of the total population. This suggests the potential for superior persistence and therefore potentially improved antitumour efficacy compared to the ConA group [46] (Figure 3a). Furthermore, the CD3/CD28 beads stimulation followed by IL-7 and IL-15 supported a significantly more efficient in vitro expansion. An approximately 5-fold greater proliferation was demonstrated in the NT group and about 3-fold more for the mA20 group (Figure 3b). However, although we harvested CAR T cells at day 5 after retroviral transduction, we demonstrated here that the fold change of proliferation did not increase after day 2 and even slightly decreased. Hence, for further research involving murine CAR T cells, it is advisable to consider decreasing the expansion time period to two to three days for an optimized procedure (Figure 3b). In summary, for T cell activation, CD3/CD28 beads outperformed ConA, while IL-7 and IL15 enabled more extensive in vitro expansion compared to IL-2.

We assessed if a single freeze-thaw (FT) cycle compromised the quality and function of murine CAR T cells. Compared with freshly harvested cells, murine CAR T cells thawed from cryopreservation (Beads-FT-mA20T) exhibited approximately 8% increased transduction efficiency, around 10% reduced PD1 expression and a predominantly CARTcm phenotype with no significant impact on cell survival (Figure 3a). We evaluated cytokine release, antigen specific cytotoxicity and phenotypic characterization to further evaluate the functionality of the Beads-FT-mA20T. More than double the IFN-γ levels were detected when co-culturing TB32043mb6s2 with Beads-FT-mA20T than with ConA-mA20T (Figure 3c), suggesting higher specific anti-tumour activity [31,47] and enhanced target adhesion that could mediate productive cytotoxicity in solid tumours [48]. Simultaneously, increased tumour-derived IL-33 was released, indicating enhanced effector function of endogenous CD8+ T cells [49], which improves CAR T cell antitumour function [50]. Importantly, a marked increase in levels of CXCL9, CXCL10 and CCL5, which are key chemokines that direct the migration of immune cells into solid tumours [51] were observed, suggesting enhanced infiltration capacity (Figure 3c) [51,52]. However, there were also elevated cytokines from the Beads-FT-mA20T that could potentially contribute to the tumour-promoting responses, as represented by increased levels of CXCL16, CXCL2 and CCL17 [53,54,55] (Figure 3c) (See Section 4). For functional validation, we examined Beads-FT-mA20T CAR T cells for cytotoxicity and phenotype. Beads-FT-mA20T maintained efficient anti-αvβ6 cytotoxicity (Figure 3d). Moreover, a consistent quality was demonstrated, with approximately 25% of Beads-FT-mA20T being CAR-expressing, predominantly CD8+ T cells, exhibiting low PD1 expression levels and primarily composed of a CARTcm cell phenotype (Figure 3e). Overall, we introduced a practical and efficient process for murine CAR T cell production and cryopreservation. This can benefit CAR T cell investigations in immunocompetent models and enable researchers to prepare CAR T cells in advance for preclinical studies requiring a substantial number of mice (Figure 2). The Beads-FT-mA20T cells were used in subsequent in vivo experiments.

### 3.3. Oncolytic Vaccinia Virus Combined with TGF-β Antibody Improves the Efficacy of CAR T Cell Therapy in an Immunocompetent Pancreatic Cancer Model

To maximise the efficacy of Beads-FT-mA20T cells we co-treated tumour-bearing mice with oncolytic vaccinia virus armed with mouse interleukin-21 (OVV-mIL21) and TGF-β-blocking antibody (anti-TGFβ). The OVV-mIL21 was developed previously and shown to exhibit antitumour potency [28]. Interleukin-21 (IL-21) has been reported to preserve a naive-like phenotype in T cells [56] and supports the effector function of cytotoxic CD8+ T cells [57]. The antitumour and local immune activation properties of OVV, together with the potential of IL-21 to promote an antitumour immune response, was chosen to enhance CAR T cell persistence and increase antitumour activity in solid tumours. Use of OVV-mIL21 alone demonstrated effective cell killing (Figure 4a) and viral replication (Figure 4b) in the wild-type TB32043 cell line, exhibiting similar antitumour activity against both TB32043 and TB32043mb6s2 (Figure 4c). The novel therapeutic regimen was studied in immunocompetent mice bearing orthotopic PDAC tumours derived from TB32043mb6s2. Described in Figure 4d, therapy was initiated one week after the orthotopic surgery, once the tumour was detectable by luminescence (Appendix A). In this study, the triple therapy regimen (CAR T + OVV-mIL21 + anti-TGFβ) included three doses of OVV-mIL21, two doses of anti-TGFβ and two doses of Beads-FT-mA20T cells. Additionally, IL-2 was administered on the same day of, and two days after, CAR T cell injection (Figure 4d). Notably, mice were pre-conditioned with CAL-101 (anti-PKC delta) prior to OVV-mIL21 treatment [58] and with cyclophosphamide prior to CAR T therapy [59] to increase therapeutic efficacy. The triple therapy strategy demonstrated better antitumour growth effects than the PBS or CAR T alone groups, with notable improvements compared to the OVV-mIL21 alone, anti-TGFβ alone and double therapy (CAR T + OVV-mIL21) groups (Figure 4e). However, severe side effects occurred in mice receiving two doses of CAR T cells. Notable weight loss was observed in groups that received CAR T cells (Figure 4f). The significant toxicity observed in the mice that received CAR T therapy necessitated their earlier euthanasia compared to those in the other groups (Figure 4g). In summary, the triple therapy demonstrated a promising antitumour effect, however, it was too toxic for the mice to tolerate.

Toxicity remains one of the biggest challenges limiting the application of CAR T cells in solid tumours. We therefore continued to explore the potential of our triple therapy by reducing the number of CAR T cell administrations from twice to once, while increasing the dosage per injection from 2 × 10^6^ to 5 × 10^6^, with the goal of minimizing the side effects of CAR T cell therapy without compromising therapeutic efficacy. First, we tested mice bearing subcutaneous TB32043mb6s2 tumours to test the tolerance of a single dose of Beads-FT-mA20T cells (Figure 4h). A safe therapy profile was observed following CAR T cell administration, with no significant adverse reactions. No significant weight loss was noted, although weight gain was affected (Figure 4i). Due to the high invasiveness and aggressiveness of TB32043mb6s2, subcutaneous tumours developed ulcerations at an early stage of the study, limiting the study duration. Only slight improvements in overall survival (Figure 4j) and tumour growth inhibition (Figure 4k) were demonstrated. Notably, the inhibitory effect of the therapy on tumour growth in the subcutaneous model was less pronounced than in the orthotopic model, suggesting the essential role of the local pancreas immune environment and the importance of restoring the TME when testing CAR T-related immunotherapeutic tools in solid tumours. However, the data suggested OVV combined with anti-TGFβ showed promise by reducing the CAR T cell dosage required, improving host safety.

### 3.4. Triple Combination Therapy Controls Orthotopic Tumour Growth and Attenuates Metastasis in Pancreatic Cancer Within an Immunocompetent Model When Extending IL-2 Supplementation

The novel triple therapy regimen described above showed promise in introducing CAR T cell therapy for pancreatic cancer, but there was still need for improvement in its efficacy. We have demonstrated that with the triple regimen, a safer profile of CAR T therapy can be achieved. Therefore, we extended the duration of IL-2 supplementation from three days to two weeks to explore potential improvements in CAR T cell function and to assess the safety profile of low-dose continuous IL-2 injection. Immunocompetent mice were implanted with orthotopic PDAC tumours using the cell line TB32043mb6s2. As in the previous protocol, three doses of OVV-mIL2 and one dose of Beads-FT-mA20T cells were administered systemically via intravenous injection, while two doses of anti-TGFβ were administered intraperitoneally. Here, we further included one group with prolonged IL-2 supplementation for 14 days after CAR T injection, while the other groups maintained the previous three-day regimen (Figure 5a).

In terms of tumour growth, triple therapy with prolonged IL-2 supplementation showed the most effective control of tumour growth within five weeks (Figure 5b), outperforming the group with 3-day IL-2 injection. However, from the fifth week, tumour growth rates increased dramatically. Notably, the CAR T alone group also had an efficient inhibitory effect on tumour growth (Figure 5b). Nevertheless, mice in the CAR T alone group were terminated at an earlier stage due to poor health, which did not allow the tumour to grow significantly (Figure 5c). Another piece of evidence presented by the tumour growth is the malignant aggressiveness and rapid progression of this model. Only the triple therapy with prolonged IL-2 demonstrated significant improvement in overall survival, highlighting the rapid progression of the tumour model was challenging our ability to fully explore the real potential of triple therapy (Figure 5c). In the PBS, CAR T and CAR T combined with OVV-mIL21 groups, starting from the fourth week after tumour implantation, mice were gradually euthanized, while the triple therapy groups demonstrated a prolonged period of no morbidity (Figure 5c). When IL-2 was administered for three days, all the mice survived until week six, whereas when IL-2 was administered for two weeks, all the mice survived until week seven (Figure 5c).

Pancreatic cancer is well-known for its early metastasis. In this study, necropsies revealed that tumour cells metastasized rapidly throughout the body, leading to ascites. The primary sites of metastasis included the spleen, liver, gut, stomach, subcutaneous tissue, lung and thoracic diaphragm (Figure 5d). The occurrence of metastasis at each site, as well as ascites, was recorded and quantified; the presence of metastasis at one of the nine sites was counted as one point, with a maximum total of nine points (Figure 5d). Excitingly, the lowest score was observed in the triple therapy groups, and consistently, the prolonged IL-2 support demonstrated the most significant effect (Figure 5d). Weight loss was observed in all treatment groups and was more severe when CAR T cells and OVV-mIL21 were part of the therapy (Figure 5e). The most severe weight loss recorded was just below 15%. After the treatment was completed, weight gradually recovered and stabilized (Figure 5e). Overall, prolonged IL-2 administration enhanced the therapeutic impact of the triple therapy and did not result in severe side effects. Improved anti-tumour efficacy was achieved with the triple therapy, particularly in inhibiting metastasis.

### 3.5. The Triple Therapy Enhances CAR T Cell Efficacy by Improving T Cell Infiltration and Persistence in Pancreatic Cancer

Failure of CAR T cell therapy in solid tumours is partially due to the immunosuppressive TME, which results in poor T cell infiltration, persistence and expansion [60]. Pathological analyses were conducted to further elucidate the effects of our triple therapy on the TME. Firstly, the expression of αvβ6 integrin was validated and found to be stable and uniform across the primary tumour and metastatic sites, including the liver, lymph nodes and gastrointestinal metastasis. Hence, this orthotopic model in immunocompetent mice can simulate and reflect human pancreatic cancer, demonstrating the potential clinical value of the observed therapeutic effects.

Next, CD3 staining was performed to evaluate T lymphocyte infiltration and local immune activation. Significantly more T cells were observed in both primary tumour and liver metastasis tissues in the triple therapy group combined with IL-2 injection for 14 days (IL-2/14d) compared to all other experimental groups (Figure 6a). For statistical validation, three metrics were used to assess T cell infiltration: the percentage of CD3+ cells, indicating the proportion of CD3+ cells relative to the total cell count in the tissue; CD3+ density, representing the average number of CD3+ cells per square millimetre (mm²) across the entire tumour area; and CD3+ distance, reflecting the average distance of CD3+ cells to the tumour border. Consistently, in the primary tumour, the triple therapy group (IL-2/14d) showed a markedly increased percentage of CD3+ cell number and CD3+ cell density, along with greater CD3+ distance from the tumour border and thus depth of infiltration (Figure 6b. A similar trend was observed in liver metastases. Notably, the importance of extended IL-2 supplementation was further highlighted when compared to the triple-therapy group, which received only three doses of IL-2. Moreover, considering the higher baseline density of CD3-positive cells present in the liver metastases across all groups (Figure 6a,b), the CD3+ density in the liver metastasis did not increase as dramatically as in the primary tumour. T cells were found to penetrate more deeply into the liver metastasis than into the tumour, as demonstrated by the dramatically increased CD3+ distance (Figure 6b), which may explain the more pronounced effect of triple therapy in controlling and targeting these metastases and potentially other metastases. Overall, the triple therapy (IL-2/14d) effectively enhanced T cell infiltration in pancreatic cancer compared to CAR T cells alone, presumably by reducing the immunosuppressive characteristics of the TME.

## 4. Discussion

Despite significant improvements in survival rates across various cancer types, PDAC remains one of the deadliest neoplasms, with a persistently low 5-year survival rate [1]. Current therapies for advanced PDAC offer limited efficacy, illustrating the urgent need for innovative therapeutic approaches [61]. In this study, we aimed to investigate a novel combinatory CAR T cell strategy in immunocompetent models, recognizing that the immune system plays a crucial role in determining the success or failure of CAR T cell therapy in solid tumours [6,62,63,64]. The immunosuppressive microenvironment in PDAC poses a significant barrier to effective CAR T cell therapy. PDAC is characterized by a dense stroma and a high infiltration of immunosuppressive cells [65,66,67,68]. The immune infiltrate within the PDAC TME includes regulatory T cells (Tregs), myeloid-derived suppressor cells (MDSCs) and tumour-associated macrophages (TAMs) [33,68,69,70,71].

Tregs, characterized by the expression of the transcription factor FOXP3, maintain immune homeostasis and prevent autoimmune reactions [72,73]. However, in PDAC, their presence within the tumour microenvironment is often substantially elevated, where they actively suppress anti-tumour immune responses. This suppression primarily occurs through the release of inhibitory cytokines, including TGF-β, IL-10 and IL-35, which dampen effector T-cell functions, thus reducing the activation, proliferation and cytokine production of the CAR T cells that manage to infiltrate the tumour [34,73,74]. This complex network of cytokines creates a formidable immunosuppressive barrier, which CAR T-cell therapies must overcome to be effective in PDAC treatment.

In order to recapitulate the complex TME of PDAC and to operate within an immunocompetent mouse model, we needed a relevant selective therapeutic target for PDAC and then to develop a murine CAR that would specifically bind to that target. The integrin αvβ6 is expressed weakly if at all in normal pancreas but is significantly upregulated in most PDAC [8,9,38,75,76]. We previously generated an αvβ6-expressing mouse PDAC cell line TB32043mb6s2 that we utilised as the tumour model in our study [16]. The αvβ6-specific peptide A20FMDV2 [14,17,77] has been used as the binding domain of an αvβ6-targeting CAR [15,20]. The A20FMDV2 CAR showed impressive anti-tumour effects in heterotopic immunodeficient PDAC models [15].

We further developed this concept to produce a fully murine CAR containing an A20FMDV2 peptide binding domain (Figure 1a). The fully murine A20FMDV2 CAR could then be used in an immunocompetent mouse model. Intravenous injection of mouse CAR T cells expressing the A20FMDV2 (mA20CART) significantly increased the survival of mice bearing orthotopic TB32043mb6s2 tumours (Figure 1h,j), the difference being slightly greater in immunodeficient NSG mice versus syngeneic C57Bl/6 mice. These results suggest that the immune system suppresses mA20CART efficacy. Our data confirmed mA20CART are αvβ6-selective in vitro and efficacious in vivo compared to non-targeting CAR T cells.

As our initial experimental design included multiple infusions of CAR T cells, we needed to develop a method that ensured we could generate large numbers of murine CAR T cells and that their quality and efficacy was the same for every injection. Figure 2 outlines the resulting production methods that allowed generation of cryopreserved stocks of efficacious αvβ6-specific CAR T cells. Notably, the analysis of cytokine release from thawed cells was higher for multiple different cytokines (Figure 3c), suggesting a more active phenotype.

When co-culturing CAR T cells with TB32043mb6s2 cells in vitro, a significant secretion of CXCL2, CXCL16 and CCL17 was observed, which are potentially tumour-promoting factors that could hinder therapeutic efficacy in vivo (Figure 3c). In PDAC, blocking CXCL2 activity has been reported with improved T cell penetration [53]. Increased CXCL16 has been associated with PDAC initiation, progression, migration and invasion. Notably, the effect of CXCL16 has been reported to be TGF-β dependent [78,79]. Furthermore, increased CCL17 has been associated with higher infiltration and tumour-promoting activity of regulatory T cells (Tregs) [80,81].

To improve the efficacy of the CAR T cell treatment in the immunocompetent model we sought to combine it with additional therapies. The combination of oncolytic viruses and CAR T has been reported previously [82,83,84,85,86,87,88]. In this study, we employed oncolytic vaccinia viruses armed with IL-21 (OVV-mIL-21) [28,29]. IL-21 serves as a potent inducer of CD8+ T cell activation [89] and promotes the maturation, activation and cytolytic potential of natural killer (NK) [90] and NKT cells [91]. Additionally, it enhances the production of tumour-specific IgG by B cells [92], inhibits FOXP3-expressing regulatory T cells (Tregs) [93] and suppresses angiogenesis [94]. Importantly, IL-21 has demonstrated a favourable safety profile in clinical trials, with no adverse effects observed even at high doses [95]. The combination of OVV equipped with IL-21 enhanced the anti-tumour activity of Beads-FT-mA20T cells and reduced metastasis (Figure 4 and Figure 5). Although the behaviour and efficacy of OVV in immunocompetent mice may not fully translate to humans, it has shown promising results and a strong safety profile in clinical trials [96,97].

TGF-β is a cytokine that plays a significant role in tumour immune evasion by suppressing immune responses, promoting tumour cell proliferation and facilitating metastasis [98,99,100,101,102,103]. High TGF-β levels in the tumour microenvironment can hinder CAR T cell activation, proliferation and cytotoxicity, thereby reducing their therapeutic potential [33,104]. Studies have demonstrated that inhibiting TGF-β, either through small molecules or genetic modifications of CAR T cells to resist TGF-β signalling, can improve CAR T cell persistence, reduce expression of exhaustion markers, enhance cytotoxic activity and improve tumour regression rates [105,106,107,108,109,110]. Furthermore, preclinical models have shown that combining CAR T cells with TGF-β inhibitors results in a more favourable immune microenvironment by decreasing regulatory T cells and myeloid-derived suppressor cells, both of which are associated with immunosuppression in tumours [111].

While TGF-β inhibition shows potential in boosting CAR T cell efficacy, it can also result in significant adverse effects due to the widespread role of TGF-β in immune regulation and tissue homeostasis [112,113]. TGF-β is critical for maintaining immune tolerance and preventing autoimmunity; therefore, its chronic inhibition cause toxicities including autoimmune-like symptoms such as colitis, dermatitis and pneumonitis, which have been reported in clinical trials involving TGF-β inhibitors [114,115,116]. Furthermore, some trials have observed an increased risk of cardiovascular issues, possibly due to altered signalling in blood vessels and the myocardium [115]. Given these potential side effects, transient TGF-β inhibition prior to CAR T cell infusion may mitigate these risks while enhancing CAR T cell efficacy.

Initial triple therapy studies showed some improvement in survival (Figure 4) but was most notable for the toxicity experienced by all mice that received double inoculum of CAR T cells, reaching their humane endpoint due to severe weight loss and sometimes ascites. We modified the treatment schedule to include only a single dose of Beads-FT-mA20T cells to reduce study-limiting toxicities that we previously saw. Additionally, we adapted the model to use mice bearing subcutaneous TB32043mb6s2 tumours, proposing that the location under the skin might reduce the very aggressive metastatic phenotype of our PDAC model and provide a longer time frame for the therapy to take effect. However, the subcutaneous tumours developed ulcers within the treatment time frame that also forced an earlier humane endpoint for the welfare of the mice. Additionally, the triple therapy in this model was less effective than when applied to the orthotopic model, which could indicate that a tissue specific tumour microenvironment while also protecting cancers due to immunosuppression, is also required to enhance therapies designed to reduce the immunosuppression (Figure 4).

We again used mice with orthotopically injected TB32043mb6s2 and applied the modified triple therapy described in Figure 5 that included a single dose of Beads-FT-mA20T cells but adding an additional arm of extended IL-2 supplementation to enhance the longevity of the CAR T cells. It was clear that the triple therapy with extended IL-2 supplementation was the only treatment to moderately, but statistically significantly, improve the survival of the mice (Figure 5c). When the grossly visible metastases were quantified, we also saw that combination of Beads-FT-mA20T and OVV-mIL21 resulted in a reduction in the metastasis compared with CART cells alone (Figure 5d). Antibody blockade of TGF-β enhanced this effect, presumably by acting on local immune cells in the TME and reducing their immunosuppressive effects.

As the design of our study was to enhance T cell infiltration into tumours, we analysed our treated tumours by immunohistochemistry. First, we confirmed that the target antigen αvβ6 remained strongly expressed in primary tumours and metastases (Figure 6a). The data show clearly that in the primary tumours there was a significant increase in infiltration of CD3 positive cells in the triple therapy, in terms of the total number of CD3+ cells per tumour and their density per um^2^ of tumour (Figure 6a,b). There was also a trend to deeper penetration into the tumour, which reached significance in the metastases (Figure 6a,b).

This study demonstrates promising clinical potential for pancreatic cancer immunotherapy. Human CAR T cells (A20-28z) targeting αvβ6 have previously been developed [15,20]. These CAR T cells incorporate the same A20FMDV2 binding domain as the mA20CAR T cells described herein and represent a strong candidate for further clinical development within the combination therapy framework established in this study. Conducted in an immunocompetent model, the study provides valuable preclinical evidence to support future clinical translation. Notably, the OVV-mIL21 applied here is currently undergoing preclinical development for an Investigational New Drug (IND) application in pancreatic cancer [117], laying a solid foundation for its future incorporation into this combination therapeutic strategy. Interestingly, we did explore the use of anti-PD-1 antibody co-therapy with our triple-therapy but discovered that anti-PD1 antibody was sufficient to abrogate TB32043mb6s2 tumour growth, eliminating our opportunity to investigate improving CAR T therapy of solid cancers.

In conclusion, our study demonstrates the therapeutic potential of combining CAR T cells, oncolytic virus OVV-mIL21 and TGF-β inhibition in an immunocompetent model of PDAC, which is one of the most challenging cancers to treat. Although translating this complex triple-combination approach to human patients presents logistical and safety challenges, our findings highlight meaningful outcomes in this aggressive PDAC model. Notably, we observed a significant reduction in metastatic spread, which can be attributed to the synergistic effects of OVV-mIL21 and CAR T cell activity, alongside a marked increase in T cell infiltration within tumour sites. This enhanced immune infiltration correlates strongly with improved overall survival, underscoring the potential of overcoming PDAC’s immunosuppressive microenvironment. Our data support the viability of CAR T cell therapy for solid tumours like PDAC, provided we can address the dual challenges of immunosuppression and associated toxicities, paving the way for further translational research in solid tumour immunotherapy.

## 5. Conclusions

The failure of CAR-T cell therapy in solid cancers is due to the host tumour microenvironment’s ability to suppress the activity and ingress of CAR-T cells into tumours, largely by the creation of a powerful immunosuppressive environment. In this study we targeted the immunosuppressive cells by antibody blockade of TGF-β and introduced the cytokine IL21 in oncolytic viruses as a chemokine ‘beacon’ to attract and support effector immune cells. The experimental design required we create a novel mouse αvβ6-targeting CAR T cells and develop methods for cryopreservation of mouse CAR T cells. This allowed us to conduct our triple-therapy (OVV+anti-TGFβ+CAR T) that resulted in improved CAR T cell penetration into pancreatic cancers. The therapy delayed the growth of primary tumours and reduced metastasis but the overall survival of mice bearing these aggressive tumours was only moderately improved. These observations still offer promising clinical insights for future therapeutic designs for improved CAR T cell therapies of solid cancers.

## Figures and Tables

**Figure 1 cancers-17-01534-f001:**
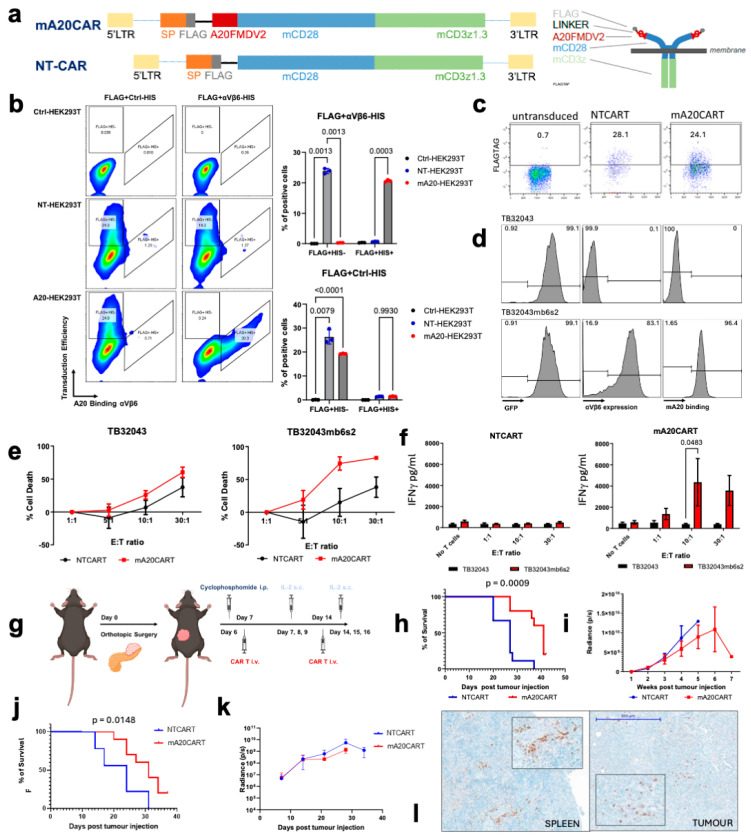
Murine CAR T cells expressing A20FMDV2 demonstrate an αvβ6-specific anti-tumour effect toward pancreatic cancer cells overexpressing integrin in vitro and in vivo. (**a**) Schematic of the murine CAR T cells with an A20FMDV2 binding domain (or no binding domain), mCD28 hinge and transmembrane domain and mCD3 z1.3. N-terminal FLAGtag used for detection of CAR expression (a more detailed image is shown in Appendix A). (**b**) Representative flow cytometry plot and quantification of the expression of the FLAGtag and αVβ6-his binding on HEK293T cells transfected with NT or mA20CAR. The transduction efficiency was demonstrated by FLAGtag positivity, while A20-αvβ6 binding was represented by anti-HIS positivity on the recombinant integrin. Statistical analysis refers to the frequency of the total (*n* = 3). A two-way ANOVA test was used to determine statistical significance. The *p* values are indicated on the figures. (**c**) Representative plot of flow cytometry to detect the N-terminal FLAG tag shows transduction of murine T cells expressing mA20 and NTCAR. (**d**) Representative histograms of flow cytometry to determine expression of GFP, αvβ6 (clone 10D5) and binding of Cy3-A20MDV2 on TB32043ctrl and TB32043mb6s2 cells. (**e**) Viability of TB32043ctrl or TB32043mb6s2s2 cells co-cultured with varying E:T ratios of mA20CART or NTCART cells for 48 h. *n* = 3. (**f**) Quantification of IFN-γ secretion of NT or mA20CART following 48-h co-culture with TB32043ctrl or TB32043mb6s2 cells at varying E:T ratios. *n* = 3; two-way ANOVA test. (**g**) Schematic illustrating experimental design of the in vivo experiment. A total of 1000 TB32043mb6s2s2 cells were orthotopically engrafted into C57BL/6 or NSG mice. On days 7 and 14, 10 × 10^6^ mA20 or NTCART cells were administered intravenously. Subcutaneous IL-2 was given on the day of CAR T cell injection and on the two days following. IVIS bioluminescence imaging was performed at various time points. (**h**) Survival curves of NSG mice. *p* < 0.05; logrank Mantel–Cox test. (**i**) Quantification of tumour radiance in the immunodeficient model over time. (**j**) Survival curves of the C57BL/6 mice. *p* < 0.05; logrank Mantel–Cox test. (**k**) Quantification of tumour radiance in the immunocompetent model over time. (**l**) Representative sections of TB32043mb6s2-derived spleen and tumour tissues from NSG mice stained for CD3 following treatment with mA20CART or NTCART cells in the immunocompetent model. Images show a low power and high-magnification insert of the cell infiltrate.

**Figure 2 cancers-17-01534-f002:**
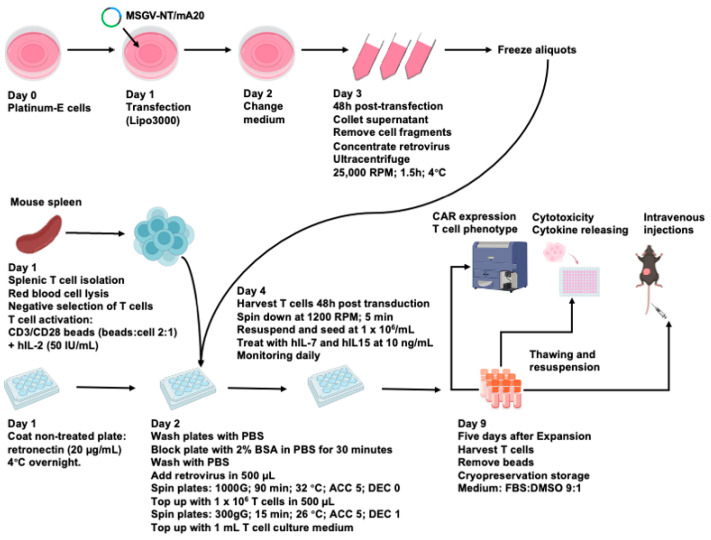
Murine CAR T cell production and cryopreservation. Prior to splenic T cell isolation, retroviral supernatant was prepared. Platinum-E cells were plated and the CAR vector was transfected using Lipofectamine 3000 (Lipo3000; Fisher Scientific). The growth medium was refreshed 24 h after to reduce the toxicity of the transfectant. After 48 h, supernatants were collected and cellular debris was removed by centrifugation. Retroviruses were concentrated and either used immediately or frozen in aliquots. Primary T cells were isolated from healthy C57BL/6 mouse spleens and activated with CD3/CD28 beads and hIL-2. For transduction, untreated plates were first coated with retronectin, followed by the addition of retroviruses and activated T cells in sequence, with centrifugation after both additions. After 48 h, transduced T cells were harvested from the plate and expanded in medium containing hIL-7 and hIL-15. Five days later, the engineered T cells were harvested, beads were removed and the expression of CAR and phenotypic characteristics were assessed by flow cytometry. The engineered T cells were used immediately or frozen overnight (10% DMSO/90%FBS) to −80 °C before transfer to liquid nitrogen. After thawing, viability was determined by flow cytometry, and the T cells were used for in vitro cytotoxicity assays or administered in vivo.

**Figure 3 cancers-17-01534-f003:**
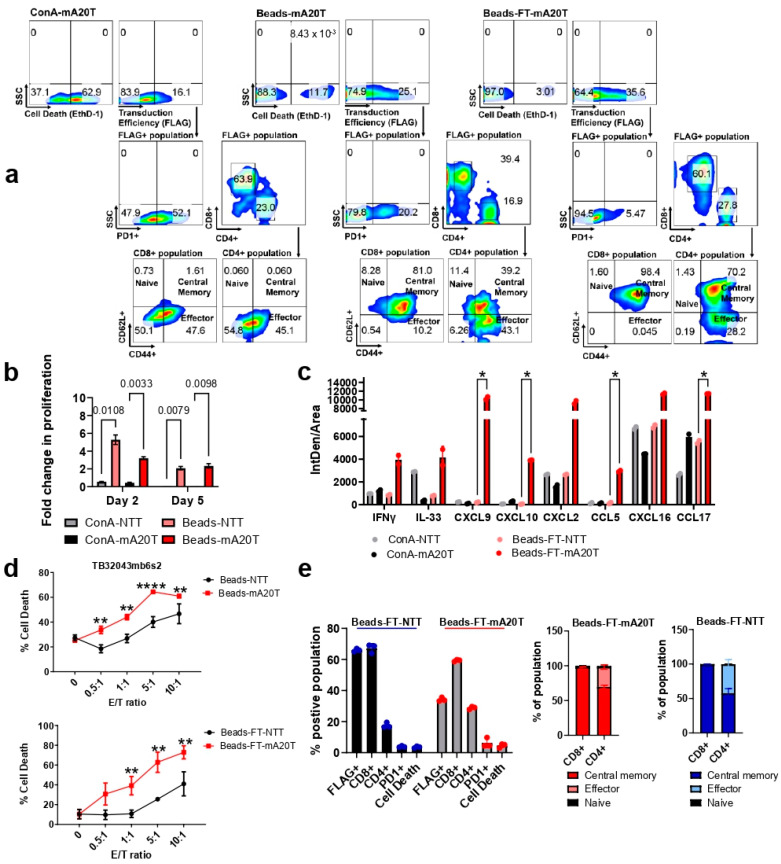
Characterisation, specificity and efficacy of the cryopreserved murine CAR T cells. (**a**) Comparison of differently activated and expanded murine CAR T cells. The numbers represent the percentage of cells expressing markers detected by flow cytometry. Cell death was detected using Ethidium Homodimer-1 (EthD-1) staining, while transduction efficacy was evaluated using anti-FLAGtag antibody. APC negative controls were applied. (**b**) Fold change in the number of viable T cells was detected using CellTiter-Glo (Promega) and normalized to day 1. (**c**) Cytokine release was assessed using the Proteome Profiler Mouse XL Cytokine Array (Bio Techne), with analysis of IntDen/Area conducted using ImageJ (version 5.3). (**d**) CAR T cell cytotoxicity assay. CAR T cells were co-cultured with TB32043mb6s2 cells for 24 h. Cell samples were collected, and cell death rates were detected by flow cytometry using EthD-1 staining. (**e**) The phenotypes of the Beads-FT-NTT and Beads-FT-mA20T cells were evaluated by flow cytometry. The numbers represent the percentage of cells expressing the marker. A two-way ANOVA test was used to determine statistical significance. * *p* < 0.05, ** *p* < 0.01, **** *p* < 0.0001. ns, not significant. Error Bars: ±SD.

**Figure 4 cancers-17-01534-f004:**
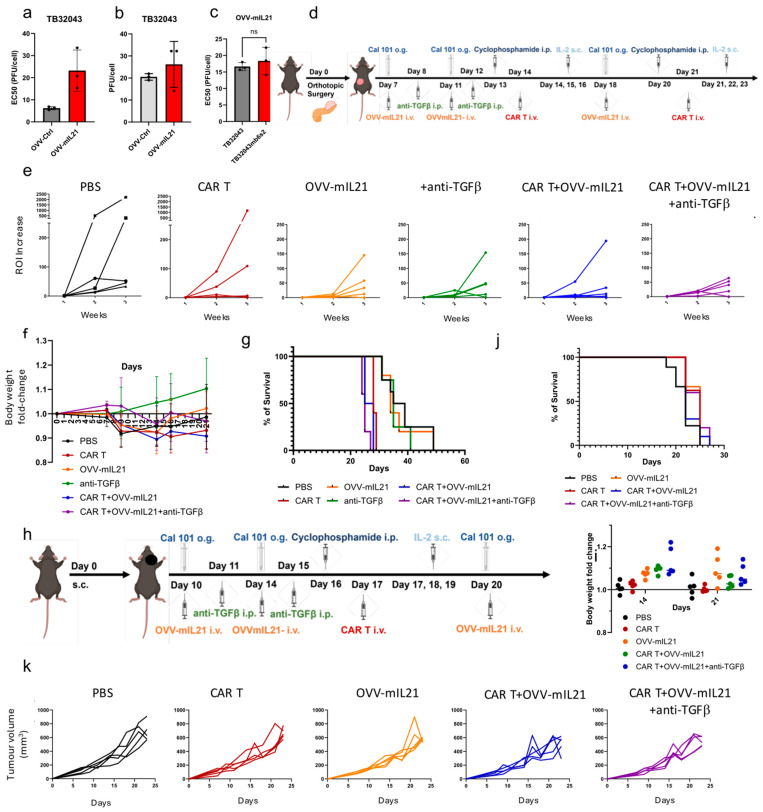
Oncolytic vaccinia virus combined with TGF-β antibody aids the efficacy of CAR T cell therapy in an immunocompetent PDAC model. (**a**) OVV-mIL21 cytotoxicity assay. Normalized EC50 values (PFUs/cell) were determined from sextuplicate MTS assays conducted 6 days after infection with a serial dilution of the virus (*n* = 3). (**b**) Viral replication assay. Viral replication was assessed by TCID50 assay on CV1 cells 48 h after virus infection at 5 PFUs/cell (*n* = 3). (**c**) Viral cytotoxicity assay (*t*-test; *n* = 3). (**d**) Schematic illustrating the experimental design. On day 0, 1000 TB32043mb6s2 cells were orthotopically engrafted into C57BL/6 mice. The mice were randomized and grouped prior to the first treatment. CAL-101 was administered at 200 µg/injection. OVV-mIL21 was given at 10^8^ plaque-forming units (PFUs)/injection for the first two injections, with the third injection at 2 × 10^8^ PFUs/injection. Cyclophosphamide was administered at 100 mg/kg. CAR T cells were administered twice at 2 × 10^6^ cells/injection. IL-2 was given on the day of and two days after CAR T cell injection at 45,000 IU/injection. Anti-TGFβ was administered at 500 µg/injection. (**e**) Tumour growth rate. All ROI values were normalized to the starting (week one) ROI values. Bioluminescence was measured once per week, and tumour growth for each mouse was plotted (*n* = 5). (**f**) Body weight fold change. All weight data were normalized to the starting weight (*n* = 5). (**g**) Overall survival curve for the orthotopic experiment. (**h**) Schematic illustrating the experimental design. TB32043mb6s2 cells were subcutaneously implanted in the right shoulder. CAR T cells were administered once at 5 × 10^6^ cells/injection. The dosages of other reagents were the same as described above. (**i**) Body weight fold change (*n* = 5). (**j**) Survival curve for the subcutaneous experiment. (**k**) Tumour volume for the subcutaneous experiment. Tumours were measured twice weekly using callipers, and the tumour size for each mouse was plotted (*n* = 5).

**Figure 5 cancers-17-01534-f005:**
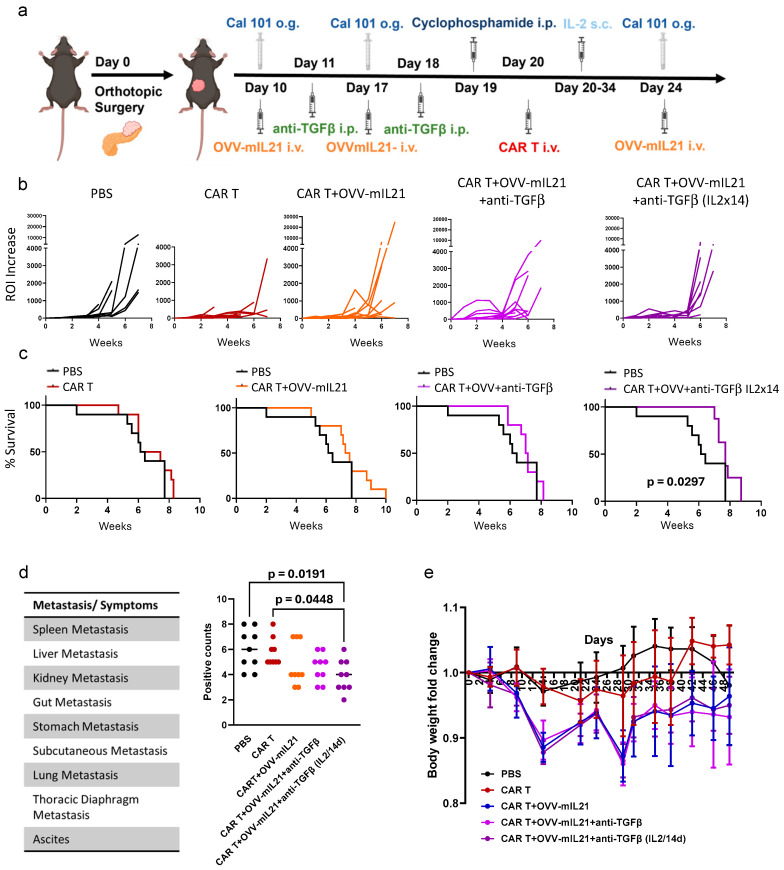
The triple therapy strategy controls tumour growth and attenuates metastasis. (**a**) Treatment regime for triple therapy with extended IL-2 injection. On Day 10, mice received CAL-101 via oral gavage (o.g.) 2–3 h before OVV-mIL21 intravenous (i.v.) injection; PBS served as a control. Twenty-four hours after virus injection, mice were administered anti-TGFβ intraperitoneally (i.p.), followed by a second dose of CAL-101 and OVV-mIL21 on Day 17, and a second anti-TGFβ dose the following day. On Day 19, mice were pre-treated with cyclophosphamide i.p., with CAR T cells injected i.v. the next day. hIL-2 was injected subcutaneously immediately after T-cell injections and daily for the subsequent 13 days. On Day 24, mice received the third dose of CAL-101 and OVV-mIL21. Dosing: OVV-mIL21: 1 × 10^8^ PFUs/injection (first two injections), 2 × 10^8^ PFUs/injection (third injection); CAR T cells: 5 × 10^6^ cells/injection; CAL-101: 200 µg/injection; cyclophosphamide: 100 mg/kg; IL-2: 45,000 IU/injection; anti-TGFβ: 500 µg/injection. (**b**) Tumour growth was monitored once a week by IVIS. *n* = 10. (**c**). Survival curve. Statistical significance was determined using the Kaplan–Meier test. *n* = 10 mice. (**d**). Positive counts represent the total number of organs with metastases and the occurrence of ascites for each mouse. Each data point represents an individual mouse. *n* = 9 mice. A two-way ANOVA test was used to determine statistical significance at 95% confidence. The *p* values are indicated on the figures. (**e**) Body weight fold change *n* = 10.

**Figure 6 cancers-17-01534-f006:**
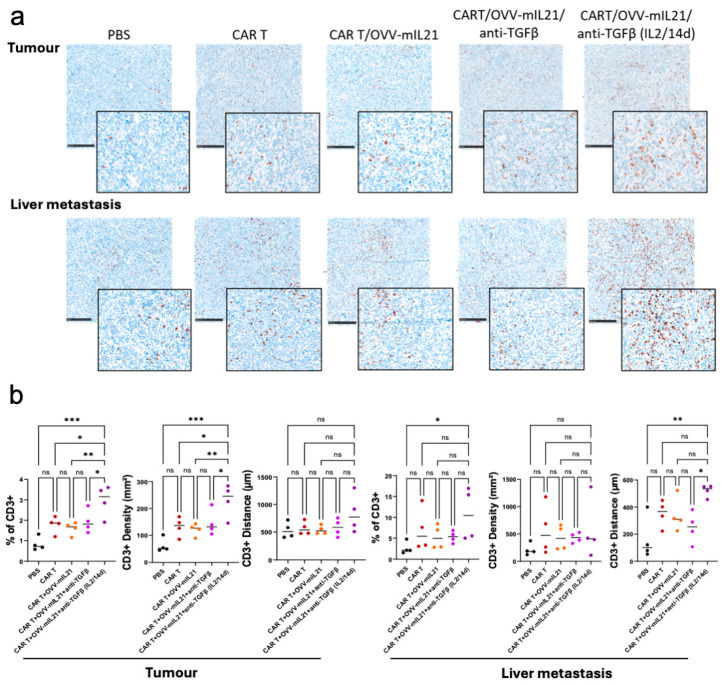
The triple therapy regimen enhances CAR T cell efficacy by improving T cell infiltration and persistence in an immunocompetent model of PDAC. (**a**) IHC images (low-power and an enlargement) demonstrating CD3-positive cells in primary tumour and liver metastasis. Scale bars represent 500 µm. (**b**) Quantification and analysis of CD3-positive cells in tumour and liver metastasis. The HALO AI image analysis platform was used for analysing the whole tissue sections for CD3 positivity. Whole-tumour spatial analysis was applied, excluding artifacts, necrotic areas, non-tumour tissue and the tumour–tissue interface. Statistical significance was determined using two-way ANOVA tests (* *p* < 0.05, ** *p* < 0.01, *** *p* < 0.001), ns = not significant.

## Data Availability

Data contained in this study are available through the corresponding authors.

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
