# Peer review of "The Combination of Oncolytic Virus and Antibody Blockade of TGF-β Enhances the Efficacy of αvβ6-Targeting CAR T Cells Against Pancreatic Cancer in an Immunocompetent Model"

_cancers, 2025, doi:10.3390/cancers17091534_

Round 1
Reviewer 1 Report
Comments and Suggestions for Authors
The Combination of Oncolytic Virus and Antibody Blockade of TGF-β Enhances the Efficacy of αvβ6-targeting CAR T Cells Against Pancreatic Cancer in an Immunocompetent Model
The manuscript presents an unconventional preclinical combination therapy that assess a novel CAR T cell therapy for pancreatic ductal adenocarcinoma (PDAC) in an immunocompetent model. This study explores a triple therapy approach combining, αvβ6-targeting CAR T cells (mA20CART), Oncolytic vaccinia virus armed with IL-21 (OVV-mIL21, TGF-β blockade in PDAC models. The research presented is very timely and relevant, given the need for more effective CAR T therapies for solid tumors. The authors have made the rationale, methods, and results in very-detailed oriented, thus making significant contributions to the field. However, there are some weaknesses that need to be addressed before considering to Cancers.
Major :
There is a serious lack of mechanistic insights into toxicity. Severe weight loss and toxicity were observed with CAR T therapy, but the cause is unclear. I would recommend authors to check whether this was this due to off-target effects or cytokine release syndrome (CRS)? An additional flow cytometry panels on immune cells (e.g., Tregs, macrophages, dendritic cells) could clarify systemic immune activation and cytokine profiles beyond IFN-γ (e.g., IL-6, TNF-α, IL-10) should be assessed to determine whether toxicity was driven by an overactive immune response.
There is very limited statistical power & sample size that I could observe. For examples, some groups had N=5, which may be too small for robust conclusions. Iwould suggest to increase the sample size to N=8-10 per group to strengthen statistical validity. Also include power analysis in the methods section to justify the chosen sample size.
The sreious flaw I could observe was inconsistent tumor growth suppression for example, the triple therapy showed good control initially, but tumors regrew after five weeks (Figure 5b).
The authors needs to investigate T-cell exhaustion markers (e.g., PD-1, LAG-3, TIM-3) by flow cytometry to determine if CAR T cells become functionally suppressed over time.
Also they need to consider adding a checkpoint blockade strategy (e.g., anti-PD-1 therapy) to sustain long-term CAR T cell function.
With current research there is a lack of direct comparison with standard therapies. The study does not compare the triple therapy to conventional PDAC treatments, such as: Gemcitabine/nab-paclitaxel, FOLFIRINOX, Anti-PD-1 or anti-CTLA-4 checkpoint inhibitors. The authors need a comparative arm with standard chemotherapy or immune checkpoint blockade which can strengthen the translational relevance of the study.
The manuscript lacks a clear discussion on how this triple therapy could be translated into clinical trials. They shall address potential manufacturing challenges for clinical-grade OVV-mIL21. Also they need to discuss whether human CAR T cells engineered against αvβ6 would require additional modifications for human trials.
Minor Issues :
The authors need to improve the clarity in Figures & Labels, The figures should include more detailed legends explaining experimental groups.
Figure 1 (CAR T cell characterization): Panel (e, f, g) need larger font sizes in axes labels.
Flow cytometry data on CAR T cell persistence in peripheral blood is missing.
Adding longitudinal analysis of CAR T cell persistence over weeks would be helpful.
IL-2 was extended from 3 days to 14 days, but why was 14 days chosen? Needs an explanation. Was IL-2 alone responsible for improving survival, or did it act synergistically with OVV-mIL21?
Regarding TGF-β Blockade: Were the effects of early vs. late TGF-β blockade evaluated? Would earlier administration of anti-TGF-β improve outcomes?
Author Response
We thank all reviewers for their incisive and helpful comments. We believe that our response to these comments has improved the manuscript and must thank them again for this.
Reviewer 1
Major:
- There is a serious lack of mechanistic insights into toxicity.
Severe weight loss and toxicity were observed with CAR T therapy, but the cause is unclear. I would recommend authors to check whether this was this due to off-target effects or cytokine release syndrome (CRS)? An additional flow cytometry panels on immune cells (e.g., Tregs, macrophages, dendritic cells) could clarify systemic immune activation and cytokine profiles beyond IFN-y (e.g., IL-6, TNF-a, IL-10) should be assessed to determine whether toxicity was driven by an overactive immune response.
We set out to determine if a triple therapy that combined CAR T cell therapy with an oncolytic virus armed with an immune stimulator and an antibody targeting the immunosuppressive cells in the TME could improve CAR T cell therapy of solid cancer. This highly ambitious goal using one the most aggressive cancers to plague humanity was partly successful in that we proved T cells entered both primary and secondary tumours in significant numbers, tumour progression was delayed and metastasis was markedly reduced. We believe these data are important for the CAR T community to read so they may encourage others to develop upon our observations. We actually agree with Reviewer 1 that it would be good to have greater mechanistic knowledge of the toxicity we observed with two doses of CAR T but it would not change the conclusions outlind above. As discussed below, we will seek to address these questions in subsequent studies that will concentrate on improving the triple-negative therapy.
- There is very limited statistical power & sample size that I could observe. For examples, some groups had N=5, which may be too small for robust conclusions. I would suggest to increase the sample size to N=8-10 per group to strengthen statistical validity. Also include power analysis in the methods section to justify the chosen sample size.
This study was to determine whether the triple therapy would show pre-clinical benefits and this required testing of mice in multiple different test groups simultaneously, with a highly complex plan requiring multiple treatments for each mouse over the several weeks. We chose the small sample size so the experiment was manageable but based on previous experience would identify those groups that would achieve statistically significant results. Our plan worked and indicated that the triple-therapy did induce significant efficacy, although not cure. Our strategy means we are now justified in future studies of specifically investigating the triple therapy alone in order to improve efficacy and addressing the mechanistic questions this reviewer has rightly suggested and thus we will use larger group sizes.
- The serious flaw I could observe was inconsistent tumor growth suppression for example, the triple therapy showed good control initially, but tumors regrew after five weeks (Figure 5b). The authors need to investigate T-cell exhaustion markers (e.g., PD-1, LAG-3, TIM-3) by flow cytometry to determine if CAR T cells become functionally suppressed over time.
With respect, describing observed experimental data as a “serious flaw” is hardly fair or reasonable. We acknowledge our triple-therapy has not performed as well as perhaps desired but we have made significant progress in overcoming aspects of why CAR T cells fail in solid cancers. Now we know which therapies work most effectively we can explore exhaustion markers and evidence of CRS in our future studies.
- Also, they need to consider adding a checkpoint blockade strategy (e.g., anti-PD-1 therapy) to sustain long-term CART cell function. With current research there is a lack of direct comparison with standard therapies. The study does not compare the triple therapy to conventional PDAC treatments, such as: Gemcitabine/nab-paclitaxel, FOLFIRINOX, Anti-PD-1 or anti-CTLA-4 checkpoint inhibitors. The authors need a comparative arm with standard chemotherapy or immune checkpoint blockade which can strengthen the translational relevance of the study.
We did investigate anti-PD-1 as a co-therapy but found that our tumour was entirely PD-1 dependent, anti-PD1 therapy preventing tumour growth and eliminating any possibility of testing our triple-therapy. We did not include these data but have mentioned our efforts in discussion. We thank the reviewer for their suggestion.
As our goal was investigating how to improve CAR T therapy we chose not to refer to standard chemotherapy of PDAC for which there are many other better studies describing their merits and failures as therapies. Additionally we could not directly compare our cellular CAR T therapy to chemotherapy with any justification.
- The manuscript lacks a clear discussion on how this triple therapy could be translated into clinical trials. They shall address potential manufacturing challenges for clinical-grade OVV-mlL21. Also, they need to discuss whether human CART cells engineered against av36 would require additional modifications for human trials.
We have added a paragraph in discussion and in conclusion that highlights what we believe our study has achieved and how this can help the CAR T research community but also encourage future human therapies may be able to emulate the improved T cell recruitment and penetration.
Minor Issues:
- The authors need to improve the clarity in Figures & Labels, the figures should include more detailed legends explaining experimental groups.
We have revised the legends.
- Figure 1 (CAR T cell characterization): Panel (e, f, g) need larger font sizes in axes labels.
We have revised font sizes
- Flow cytometry data on CAR T cell persistence in peripheral blood is missing.
CAR T cell persistence is inferred from the data showing extended IL2 treatment was associated with improved therapy.
- Adding longitudinal analysis of CAR T cell persistence over weeks would be helpful.
In future studies where we are not comparing multiple different treatment groups we intend to do this
- IL-2 was extended from 3 days to 14 days, but why was 14 days chosen? Needs an explanation.
14 days was chosen as the health of mice was diminishing in the control groups in this time frame.
- Was IL-2 alone responsible for improving survival, or did it act synergistically with OVV-mIL21?
We cannot answer this question at this time. As mentioned above, the difficulties in managing multiple different treatment groups limited the number additional studies that could be done on individual mice. In the future when we examine triple therapy in more depth we will be sure to explore this excellent suggestion.
- Regarding TGF-β Blockade: Were the effects of early vs. late TGF-β blockade evaluated? Would earlier administration of anti-TGF-β improve outcomes?
Again, we cannot address the excellent question but will explore it in future studies.
Reviewer 2 Report
Comments and Suggestions for Authors
Zhao et al developed a murine model of CAR T cells against pancreatic ductal carcinoma against αvβ6 integrin (ITGAV) which is expressed abundantly in pancreatic and other cancers, but very less in healthy cells. The oncolytic vaccinia virus peptide ( a ligand of this integrin) and IL21 carrying CAR T cells is not very effective when treated monotherapeutically but its efficacy is largely enhanced with a TGFβ blocking antibody and IL2 supplementation. Authors speculate that combination of these triple therapies would have potential treatment for PDAC.
CAR-T cell therapy, although effective in certain blood cancers but not very successful for solid tumors. Moreover, PDAC is one of the hardest treatable cancers. In this respect, the development of new treatment even with combination events has enormous significance. Authors here did considerable work to develop these potential treatments that could be translated to human PDAC.
Although paper is written well but mainly the figure representations are of poor quality. Here are some concerns that need to be addressed.
- In the author list, equal contributions are not designated.
- In materials and methods, authors should define what are TB32043ctrl and TB32043mb6s2 cells?
- The writings within the figure 1b, 1c, 1d, 1g and 1j are hardly legible. Authors must increase the font and resolution. This is an online only journal, so what are the justifications of compressing the figures so much?
- In line 154, 219, 438 and 439, authors refe3rred to supplementary materials but I did not find any supplementary materials in the m,anuscript.
- The writing within figure 3a should be in higher font and resolution.
- Line 468-470, authors should provide the number of cells/mice wt was considered as low dose that are effective without much side effect. Authors should mention the time differences of tumor remission between high doses (no of cells/mice wt) and low doses?
- The writings within figure 4d and 4h should be improved.
- The writings within figure 5a should be improved.
- In materials and methods (line 200-203), authors mentioned that they made an incision to put designed CAR T cells into the head of pancreas. How authors would put them in human pancreas?
Author Response
- In the author list, equal contributions are not designated.
This has been amended
- In materials and methods, authors should define what are TB32043ctrl and TB32043mb6s2 cells? This has been amended
- The writings within the figure 1b, 1c, 1d, 1g and 1j are hardly legible. Authors must increase the font and resolution. This is an online only journal, so what are the justifications of compressing the figures so much? This has been amended
- In line 154, 219, 438 and 439, authors referred to supplementary materials but I did not find any supplementary materials in the manuscript. Supplementary results are attached
- The writing within figure 3a should be in higher font and resolution. This has been amended
- Line 468-470, authors should provide the number of cells/mice wt was considered as low dose that are effective without much side effect. Authors should mention the time differences of tumor remission between high doses (no of cells/mice wt) and low doses? This has been amended
- The writings within figure 4d and 4h should be improved. This has been amended
- The writings within figure 5a should be improved. This has been amended
- In materials and methods (line 200-203), authors mentioned that they made an incision to put designed CAR T cells into the head of pancreas. How authors would put them in human pancreas?
This is a misunderstanding. We injected the tumour cells into the head of the pancreas not the CAR T cells.
Reviewer 3 Report
Comments and Suggestions for Authors
In this study, Zhao et al. developed a triple combination strategy to enhance the efficacy of anti-integrin aVb6 CAR T cells. The results contain sufficient information and well support the conclusion.
Major:
- How the clone of anti-aVb6 scFv is derived? Is that from previous screening or literature or patent? Please indicate the resource and experimental data if any.
- It seems that the cryopreservation of mouse CAR T did not show any advantages compared to fresh CAR T cells. I would recommend compare fresh CAR T and freeze-thawed CAR T cells in coculture assay to determine its killing ability changes.
- Have authors compared commercially available serum-free freezing media (eg. Stem Cell CS10) with traditional FBS+10%DMSO? We used to freeze mouse CAR T cells but have very poor activity after thawing. It will be very helpful if there is a method to preserve mouse CAR T cells.
- The oncolytic virus groups appears to show potent efficacy and more safety than the triple combination group. Since CAR T cells are targeting at integrin, which is widely expressed in not only tumor but normal tissues, its off-tumor killing will be a large concern. If virus can kill the tumor, it is not necessarily to be combined with CAR T cell.
Minor:
- Axis tile is missing in Fig 1c. And somehow some panels of Fig 1 are illusive, and some are clear. Please check the resolution of figures and increase the font size of flow data.
- The introduction can be refined.
Author Response
Major:
- How the clone of anti-aVb6 scFv is derived? Is that from previous screening or literature or patent? Please indicate the resource and experimental data if any.
We did not use a scFv for our CAR T. We added the peptide called A20FMDV2 (described in our review in reference [14]) which binds integrin avb6 with very high specificity.
- It seems that the cryopreservation of mouse CAR T did not show any advantages compared to fresh CART cells. I would recommend compare fresh CART and freeze-thawed CART cells in coculture assay to determine its killing ability changes.
Figure 3d demonstrates the comparison between fresh and freeze-thawed CAR T cells in terms of cytotoxicity. The freeze-thaw cycle did not affect either the cytotoxicity or specificity of CAR T cells. Excitingly, a slightly better specificity can be achieved after the freeze-thaw cycle, with reduced non-specific killing. For detailed phenotypic improvements, please refer to Figure 3a. The benefit of cryopreservation was the ability to bank large stocks of CAR T cells, that could be assessed for functionality so that it was practical to undertake complex multi-group analyses over long periods involving many mice. Such studies are almost impossible if you also have to prepare fresh mouse CAR T cells that must be ready for a specific day, in sufficient numbers and perform with the similar efficacy as a previous batch.
- Have authors compared commercially available serum-free freezing media (eg. Stem Cell CS10) with traditional FBS+10%DMSO? We used to freeze mouse CAR T cells but have very poor activity after thawing. It will be very helpful if there is a method to preserve mouse CAR T cells.
We did not compare our freezing medium with commercial serum-free freezing media in this study as our routine medium (10% DMSO and 90% FBS, as described in the paper) was sufficient to maintain sufficient viability for their efficacy in vitro and in vivo. However, it may be interesting to compare both media to further improve cryopreservation efficiency.
- The oncolytic virus groups appear to show potent efficacy and more safety than the triple combination group. Since CAR T cells are targeting at integrin, which is widely expressed in not only tumor but normal tissues, its off-tumor killing will be a large concern. If virus can kill the tumor, it is not necessarily to be combined with CAR T cell.
The expression of integrin avb6 is at low and often undetectable levels in most normal tissues, including the pancreas. We have published that targeting pancreatic cancer with the avb6 targeting peptide A20FMDV2 linked to a toxic drug, is effective and the dose not cause damage to those tissues where avb6 is expressed in detectable levels on the normal tissue (ie the stomach principally but parts of the GI and lungs-see reference [17] by Moore et al.). Thus on-target off-tumour effects of CAR T were not expected to be significant. However we do not ignore these possibilities and will explore these organs in subsequent studies to examine in CAR T cells are recruited.
The oncolytic virus applied in this study demonstrated some therapeutic effects; however, it was not sufficient to significantly reduce metastasis or enhance T cell infiltration in the tumour (Figure 5, 6). Here, we leveraged the advantages of the oncolytic virus to reshape the immunosuppressive tumour microenvironment, aiming to achieve a stronger antitumor immune response. The main objective of this study was achieved—enhancing T cell infiltration and reducing metastasis.
This work provides evidence supports further exploration of this combination approach.
Minor
- Axis tile is missing in Fig 1c. And somehow some panels of Fig 1 are illusive, and some are clear. Please check the resolution of figures and increase the font size of flow data. This has been amended
- The introduction can be refined. We have modified the introduction text